



**The Orbiting Carbon Observatory-2: First 18 months of Science Data Products**
**Annmarie Eldering[1], Chris W. O'Dell[2], Paul O. Wennberg[3], David Crisp[1], Michael R. Gunson[1], Camille Viatte[3],**
**Charles Avis[1], Amy Braverman[1], Rebecca Castano[1], Albert Chang[1], Lars Chapsky[1], Cecilia Cheng[1], Brian**
**Connor, Lan Dang[1], Gary Doran[1], Brendan Fisher[1], Christian Frankenberg[1,2], Dejian Fu[1], Robert Granat[1],**
**Jonathan Hobbs[1], Richard A. M. Lee[1], Lukas Mandrake[1], James McDuffie[1], Charles E. Miller[1], Vicky Myers[1],**
**Vijay Natraj[1], Denis O'Brien[2], Gregory B. Osterman[1], Fabiano Oyafuso[1], Vivienne H. Payne[1], Harold R.**
**Pollock[1], Igor Polonsky[3,4], Coleen M. Roehl[3], Robert Rosenberg[1], Florian Schwandner[1], Mike Smyth[1], Vivian**
**Tang[1], Thomas E. Taylor[2], Cathy To[1], Debra Wunch[3], Jan Yoshimizu[1]**
[1]{Jet Propulsion Laboratory, California Institute of Technology, Pasadena, CA}
[2]{Cooperative Institute for Research in the Atmosphere, Colorado State University, Fort Collins, CO}
[3]{Department of Geology and Planetary Sciences, California Institute of Technology, Pasadena, CA}
[4]{currently at Atmospheric and Environmental Research, Inc., Lexington, MA}
Correspondence to: A. Eldering (Annmarie.Eldering@jpl.nasa.gov)
**Abstract**
The Orbiting Carbon Observatory-2 (OCO-2) is the first National Aeronautics and Space Administration (NASA) satellite
designed to measure atmospheric carbon dioxide ($CO_2$) with the accuracy, resolution, and coverage needed to quantify $CO_2$
fluxes (sources and sinks) on regional scales. OCO-2 was successfully launched on 2 July 2014, and joined the 705 km
Afternoon Constellation on 3 August 2014. On monthly time scales, 7 to 12% of these measurements are sufficiently cloud and
aerosol free to yield estimates of the column-averaged atmospheric $CO_2$ dry air mole fraction, $X_{CO2}$, that pass all quality tests.
During the first year of operations, the observing strategy, instrument calibration, and retrieval algorithm were optimized to
improve both the data yield and the accuracy of the products. With these changes, global maps of $X_{CO2}$ derived from the OCO-2
data are revealing some of the most robust features of the atmospheric carbon cycle. This includes $X_{CO2}$ enhancements co-located
with intense fossil fuel emissions in eastern U.S. and eastern China, which are most obvious between October and December,
when the north-south $X_{CO2}$ gradient is small. Enhanced $X_{CO2}$ coincident with biomass burning in the Amazon, central Africa, and
Indonesia is also evident in this season. In May and June, when the north-south $X_{CO2}$ gradient is largest, these sources are less
apparent in global maps. During this part of the year, OCO-2 maps show a more than 10 ppm reduction in $X_{CO2}$ across the
northern hemisphere, as photosynthesis by the land biosphere rapidly absorbs $CO_2$. As the carbon cycle science community
continues to analyze these OCO-2 data, information on regional-scale sources (emitters) and sinks (absorbers) which impart $X_{CO2}$
changes on the order of 1 ppm, as well as far more subtle features, will emerge from this high resolution, global data set.



## 1 Introduction

Human activities including fossil fuel combustion, cement production, and deforestation are now adding almost 40 billion tons of carbon dioxide ($CO_2$) to the atmosphere each year (c.f. Le Quéré et al., 2015). If all of this $CO_2$ remained in the atmosphere, the atmospheric $CO_2$ concentration would increase by more than one percent (1%) per year. Interestingly, precise measurements collected by a growing global network of greenhouse gas monitoring stations over the past 60 years indicate that less than half of this $CO_2$ remains airborne (Dlugokencky and Tans, 2015) The rest is being absorbed by the oceans and the land biosphere. Measurements of the partial pressure of $CO_2$ in seawater collected over this period indicate that almost a quarter of the $CO_2$ emitted by human activities is being absorbed by the ocean (c.f. Takahashi et al. 2009), where it contributes to ocean acidification. For mass balance, another 10 billion tons of $CO_2$ must be absorbed by processes on land, whose identity and location are less well understood. Some studies have attributed this absorption to tropical (Schimel et al., 2015), or Eurasian temperate (Reuter et al., 2014) forests, while others indicate that these areas are just as likely to be net sources as net sinks of $CO_2$ (Chevallier et al., 2014). The efficiency of these natural land and ocean sinks also appears to vary dramatically from year to year (Le Quéré et al., 2015). Some years, they absorb $CO_2$ equivalent to almost all of that emitted by human activities, while in other years, they absorb very little. Because the identity, location, and processes controlling these natural sinks are not well constrained, it is not clear that they will continue to reduce the rate of atmospheric $CO_2$ buildup by half in the future (Schimel et al., 2015). This introduces a major source of uncertainty in predictions of the rate of future $CO_2$ increases, and their effect on the climate (Friedlingstein et al., 2006; Arora et al., 2013).

Measurements from the network of ground-based greenhouse gas stations accurately track the global atmospheric $CO_2$ budget and its trends. Remote sensing of the column-averaged $CO_2$ dry air mole fraction ($X_{CO2}$) from space is intended to provide finer spatial coverage enabling smaller scale *sources* emitting $CO_2$ into the atmosphere and natural *sinks* absorbing this gas at the Earth's surface to be better quantified. Surface weighted $X_{CO2}$ estimates can be retrieved from high resolution spectroscopic observations of reflected sunlight in near infrared $CO_2$ and $O_2$ bands (c.f. Rayner and O'Brien, 2001; Crisp et al., 2004; Buchwitz et al., 2006; O'Dell et al., 2012). This is a challenging space-based remote sensing observation because even the largest regional $CO_2$ sources and sinks produce changes in the background $X_{CO2}$ distribution no larger than 2%, and most are smaller than 0.25% (1 part per million (ppm) out of the background 400 ppm) (c.f. Miller et al., 2007).

The European Space Agency (ESA) EnviSat SCanning Imaging Absorption SpectroMeter for Atmospheric CHartographY (SCIAMACHY) (Burrows et al., 1995) and Japanese Greenhouse Gases Observing Satellite (GOSAT) Thermal And Near infrared Sensor for carbon Observation Fourier Transform Spectrometer (TANSO-FTS) (Nakajima et al., 2010) were the first satellite instruments designed to exploit this measurement approach. SCIAMACHY enabled retrieval of column averaged $CO_2$ and methane ($X_{CH4}$) measurements over the sunlit hemisphere from 2002 to 2012. Spectra from TANSO-FTS have been used to produce $X_{CO2}$ and $X_{CH4}$ observations since April 2009. These data have provided an important proof of concept, and are beginning to yield new insights into the carbon cycle (Feng et al., 2016; Guerlet et al., 2013; Wunch et al., 2013; Schneising et al., 2014), but improvements in sensitivity, resolution, and coverage are still needed.

The Orbiting Carbon Observatory–2 (OCO-2) is the first NASA satellite designed to measure atmospheric $CO_2$ with the accuracy, resolution, and coverage needed to detect $CO_2$ sources and sinks on regional scales over the globe. OCO-2 is a replacement for the Orbiting Carbon Observatory (Crisp et al., 2004, 2008) which was lost in 2009, when its launch vehicle malfunctioned and failed to reach orbit. OCO-2 was successfully launched from Vandenberg Air Force Base in California on July 2, 2014. Since September 6[th] of 2014, this instrument has been routinely returning almost one million soundings each day over the sunlit hemisphere. Optically thick clouds and aerosols preclude observations of the full atmospheric column, but 7 to



12% of these soundings are sufficiently cloud free to yield full-column estimates of $X_{CO2}$ with single-sounding random errors
between 0.5 and 1 ppm at solar zenith angles as large as 70 degrees.
Here we provide a brief introduction to the instrument and the mission operations to date, highlighting the global coverage,
resolution, and precision of the dataset. We describe the overall flow of data in section 4 and some key results in terms of data
quantity, quality, and features, with discussions of $X_{CO2}$ (section 4.3.1) , data quality indicators (section 4.3.3 and 4.3.4), and
overall data density (section 4.3.5). The trends in $X_{CO2}$ in space and time as seen from OCO-2 are discussed in section 5. This
paper is one of a number of papers describing the OCO-2 mission and its early results. On-orbit calibration and validation of the
Level-1 radiances are described in Crisp et al. (2016ab). Details of the $X_{CO2}$ retrieval algorithm, including filtering and bias
correction, are given in O'Dell et al. (2016), while the validation of $X_{CO2}$ via comparisons to the TCCON network are given in
Wunch et al. (2016). Finally, analysis of the Solar-Induced Fluorescence (SIF) product derived from OCO-2's Oxygen-A band is
described in Sun et al. (2016). Interested readers are advised to consult these references for details.
**2    The instrument**
The instrument of OCO-2 is a three-band spectrometer, which measures reflected sunlight in three separate bands. The oxygen
A-band (ABO2) measures absorption by molecular oxygen near 0.76 $\mu$m, while two carbon dioxide bands, labeled here as the
weak and strong $CO_2$ bands (WCO2 and SCO2 hereafter), are located near 1.6 and 2.0 $\mu$m, respectively. The instrument has 1016
spectral elements in each band, and 160 pixels are averaged in groups of ~20 along the slit, creating eight spatial footprints. The
instrument field of view creates footprints that are nominally 1.25 km in width, and the spacecraft motion spans ~2.4 km of the
ground in the 0.33 seconds of integration time. The spacecraft rotates along the orbit, maintaining a constant angle between the
plane defined by the instrument, the point observed on the ground, and the sun. As a result, the footprint shapes change during
the orbit, from nearly rectangular to very narrow swaths (see details in Crisp et al., 2016b). The rate of data collection results in
approximately 1 million sets of 3 band measurements per day.
The OCO-2 instrument collects data over very narrow spectral ranges, with a resolving power ($\lambda/\Delta\lambda$) of roughly 19,000:1 in each
band that reveals the trace gas spectral absorption lines. The spectral ranges for the $O_2$-A band, weak $CO_2$, and strong $CO_2$ are
0.7576 to 0.7726 microns, 1.5906 to 1.6218 microns, and 2.0431 to 2.0834 microns, respectively. Details of the spectral and
radiometric calibration of the instrument are reported in Lee et al. (2016) and Rosenberg et al. (2016), respectively. On-orbit
instrument performance is described in detail in Crisp et al. (2016a). Coincident measurements from the three channels are
combined into "soundings" that are analysed with a "full-physics" retrieval algorithm to yield estimates of $X_{CO2}$ and other
geophysical quantities (c.f. Boesch et al., 2006, 2011; O'Dell et al., 2012, 2016; Crisp et al., 2012).


**3    The observatory in space**
The OCO-2 observatory was launched successfully from Vandenberg Air Force Base in California on July $2^{nd}$, 2014 at 2:56 am
Pacific Daylight Time. During the 10 days following launch, the spacecraft team completed a functional check of both the
observatory and the instrument. The observatory was then maneuvered into its position in the 705 km Afternoon Constellation,
also called the A-train, arriving on August $3^{rd}$, 2014. A number of atmospheric remote-sensing satellites fly in coordination in





this constellation, such as the Moderate Resolution Imaging Spectrometer (MODIS) and Cloud-Aerosol Lidar with Orthogonal
Polarization (CALIOP) which can be used for cross comparisons of clouds and radiances. After achieving the operational orbit,
the instrument and focal planes were brought to and stabilized at their operational temperatures. During the more extensive in-
orbit-checkout (IOC) of the instrument, measurements were collected to refine the geometric, radiometric, and spectral
calibration. On August 6, 2014, the first spectral data were collected with the instrument at operating temperatures, and processed
with calibration parameters from pre-launch calibration experiments. As reported in Basilio et al. (2014) this data showed high
resolution with high signal to noise characteristics similar to the prelaunch measurements. Another critical activity during the
IOC were lunar measurements that were used, in combination with data from coastal crossings, to determine the alignment of the
spectrometers and derive the updated pointing coefficients. Calibration data collected during IOC were used to update the
instrument gain coefficients, dark correction, and to update the map of bad pixels on the focal plane. This was completed on
September 5, 2014. Data after that date is considered scientifically usable, as the instrument temperatures were stable, and the
key radiometric parameters were up to date. The OCO-2 mission formally ended the IOC period on October 12, 2014.
As of the summer of 2016, the instrument and spacecraft are performing extremely well, and data collection continues. Crisp et
al. (2016ab) provides details of data interruptions, which have been primarily driven by instrument operations.

## 3.1   The observing strategy

The observing strategy of the OCO-2 mission evolved over the first year. Initially, the strategy was to collect 16 days of nadir
data, collecting data by measuring directly below the spacecraft, followed by 16 days of glint measurements, where the
instrument is pointed towards the glint spot, to collect higher signal ocean data. This strategy was updated over time, and is
illustrated in Figure 1. The key changes were 1) the geometry of glint measurements, 2) changes to the frequency of alternating
glint and nadir mode orbits, 3) changes to the geometry of nadir orbits, and 4) the specification of some orbit paths as perpetual
glint measurements.
During early instrument checkout (Aug 7, 2014), the nominal 16-day nadir/glint pattern was disrupted after very high signals
were observed during glint measurements. For the safety of the instrument, the observing mode was shifted to nadir
measurements while the cause was investigated. We concluded that an incident of glint measurements over very still water, that
may have had a layer of highly reflective material on its surface, was the cause of the high signal measurements (see Crisp et al.
(2016b) for more discussion), and they posed no risk to the instrument, so glint data collection was restarted on September 8,
2014. In mid-September 2014 it was recognized that the measurements were consistent with a polarization sensitivity that was
rotated by 90 degrees from our expectations (again, see Crisp et al. (2016b)).  To improve the signal to noise of the glint mode
observations, particularly near the Brewster's angle, the spacecraft was yawed 30 degree during glint measurements after October
26, 2014. To provide more uniform temporal distribution of glint measurements over ocean, an additional change was made to
the data collection beginning July 3, 2015.  The nadir and glint data collection were changed to an orbit by orbit interleaving (one
orbit nadir, one orbit glint, *ad infinitum*). Over a 32-day period, nadir and glint data are collected over the same set of locations as
in the original 16 day alternating scheme, but the new approach does not have large time gaps in ocean data collection. In late
October 2015, to reduce the temperature changes of the instrument when changing from glint to nadir, the nadir geometry was
updated to collect data at the same 30-degree yaw as glint data are collected in. This allows for the collection of 3 to 5 glint orbits
in a row between nadir orbits. With this change, orbits that are solely over water, such as the Pacific and Atlantic, can be
measured in glint at all times. This type of data collection was started on November 12, 2015, and it is expected that this





approach will be used for the remainder of the mission. Figure 1 provides a calendar view of the observing strategy and data
outages.

## 4  Overall data flow

The overall flow of the data pipeline is illustrated in Figure 2. All data products except the so called 'Lite files' contain one
granule of data, which is restricted to one mode (such as nadir, glint, target, or transition). A granule corresponds to a complete
orbit of measurements except in the cases where the orbit includes a switch to target measurements. In these cases there are
separate data product files for the target and the transition before and after the target. The data that are processed as they are
collected are referred to as v7, or the forward processing stream. They use calibration coefficients that are predicted based on
recent measurements. This dataset is created in the Science Data Operations System (SDOS) at JPL. The v7r refers to the
retrospective data, or data processed with calibration coefficients based on measurements before, during, and after the
measurement time period. This dataset is typically processed on supercomputer resources (NASA's Pleiades and cloud
computing resources).
The raw (L1a) measurements are geolocated, and the calibration coefficients are applied to generate geolocated, calibrated
radiances (L1b) as discussed in Crisp et al. (2016a). These data are then passed to the preprocessors, which are used to identify
the scenes that are most likely to be cloud free and successful in generating converged retrievals. One preprocessor routine also
provides estimates of solar induced fluorescence (SIF). The $X_{CO2}$ retrievals are performed on a subset of data selected by the
preprocessors outcomes. The v7 and v7r standard (L2Std) and diagnostic (L2Dia) products report these data, which include the
$X_{CO2}$ estimates. In a final step, a bias correction and data quality flag (warn level) are integrated, and each day of quality data is
packaged into a single so-called 'Lite file' (further details in Section 4.3, and in Mandrake et al., 2015).  All L1B, L2 and Lite
products are delivered to the NASA Goddard Earth Science Data and Information Services Center (GES DISC) for distribution
and archiving (http://disc.sci.gsfc.nasa.gov/OCO-2). The L1 and L2 products are described in greater detail in the OCO-2 Data
Product User's Guide and the L1B and L2 Algorithm Theoretical Basis Documents (ATBDs) and other documents, which are
posted along with the products at the GES DISC (http://disc.sci.gsfc.nasa.gov/OCO-2/documentation/oco-2-v7) (Osterman et al.,
2015; Crisp et al., 2014; Eldering et al., 2015; Mandrake et al., 2015).

## 4.1  Calibrated radiances

The "Level 1B" (L1B) product consists of full orbits or fractions of orbits of calibrated and geolocated spectral radiances from
the ABO2, WCO2, and SCO2 channels. The details of the transformation of raw measurements into calibrated spectral radiances
are discussed in the L1b Algorithm Theoretical Basis document (Eldering et al., 2015). The pre-flight spectral and radiometric
calibration are discussed in Lee et al. (2016) and Rosenberg et al. (2016). The in-flight performance is discussed in detail in Crisp
et al. (2016a). The L2 data products are not impacted by the calibration issues discussed in Crisp et al. (2016a) with the exception
of time dependent radiometric correction factors that are now understood to be in error for the v7/v7r data, with an increasing
error in time. This radiometric error has a magnitude of about 4% by 18 months into the mission. Analysis shows that a radiance
error of 4% will impart an $X_{CO2}$ error of 0.22 ppm, 0.12 ppm, and 0.4 ppm in nadir land, glint land, and glint water
measurements, respectively. This error is not addressed in the analysis presented here, data are used as provided in the v7/v7r
files.





## 4.2 Preprocessors

For the v7 and v7r OCO-2 dataset, the A-band (ABP) (Taylor et al., 2016) and IMAP-DOAS (IDP) preprocessors (Frankenberg
et al., 2011, 2012, 2014) were used for the selection of data to be processed to L2. To limit the demands on the computing
system, no more than 6% of data collected each day are processed to L2 in the v7 forward processing stream. The v7r processing
stream includes all data that meets pre-processing criteria, which is on average 17.9 % for glint data and 6.6% for nadir. Taylor et
al (2016) describes the preprocessor outcomes in detail. In summary, the A-band preprocessor compares the measured radiance
spectra with spectra calculated with a non-scattering forward model to test for the presence of clouds. The IDP also uses a non-
scattering forward model, but it is applied to the WCO2 and SCO2 bands independently. Ratios of the single band column
retrievals are then analyzed to identify scenes that are impacted by clouds and aerosols. As reported in Taylor et al (2016) the
combined ABP and IDP OCO-2 preprocessors screen approximately 85-90% of the co-located data that MODIS reports to be
cloudy, with overall global agreement of ~85% between the two sensors. The regions of significant disagreement were found to
be tropical and subtropical oceans and desert land. Comparisons to CALIOP measurement of the vertical distribution of cloud
optical thickness confirmed the conclusion derived from simulations that the combined ABP and IDP preprocessors successfully
identify high, optically thin clouds and midlevel clouds and aerosols, but fail to identify contamination in about 25% of the cases
of low, optically thick clouds and aerosols. Additional pre-filters remove all land data south of 65S, and further limit the surface
albedo in the $O_2$-A band to less than 0.55 for a rough proxy of the presence of snow and ice on the ground, which can cause the
retrievals significant problems (O'Dell et al., 2012).

## 4.3 Level 2 algorithm products

The OCO-2 project reports two key products at L2 (derived geophysical data at the spatial resolution of the measurement), the
dry air mole fraction of carbon dioxide ($X_{CO2}$) and solar induced fluorescence (SIF). As described in the preprocessor section,
only a subset of data are considered to be sufficiently cloud (and aerosol)-free (optical depths less than ~0.35) for the next step of
processing in the L2 Full Physics algorithm, which produced the $X_{CO2}$ data product. The SIF product is generated by the IDP
preprocessors (Frankenberg et al., 2014). As described in Frankenberg et al., (2014), most of the fluorescence signal is retained,
even through moderate clouds (optical depths up to 5). As a consequence, SIF results are reported for a much larger fraction of
the OCO-2 observations compared to the $X_{CO2}$ product.
The OCO-2 retrievals for $X_{CO2}$ are created using the full physics algorithm that has been described previously (O'Dell, et al.,
2012; O'Dell et al., 2016). The retrieval algorithm is based on an optimal estimation scheme and an efficient radiative transfer
technique that accounts for multiple scattering and polarization effects. A standard cost function is minimized to find the state
vector that produces the maximum *a posteriori* probability. While the focus is the retrieval of $X_{CO2}$, other parameters such as
surface albedo, aerosols, temperature, water vapor, and wind speed (for water surfaces only), are co-retrieved. Prior to the launch
of OCO-2, this algorithm was adapted for application to the GOSAT measurements, with these results reported in O'Dell et al.
(2012) and Crisp et al. (2012), and for OCO-2 it remains largely unchanged from what was reported in those papers.
The $X_{CO2}$ data are reported in the L2_Standard files and the L2_Diagnostic files, where the diagnostic files contain additional
information that may be useful for detailed assessment of the algorithm and for the modeling community (Osterman et al., 2015).
Examples of the additional information are the averaging kernels and the *a posteriori* covariance matrix, $\hat{S}$. In v7, the L2



Standard and Diagnostic files, containing about 60,000 soundings per file, do not contain warn levels values which indicate data
quality (Mandrake et al., 2013), nor has a bias correction been applied.  This information is calculated subsequently and included
in the Lite files described below.
A summary daily data product, referred to as the Lite files, is created, to simplify data volumes and data structures. Specific files
for $X_{CO2}$ (Mandrake et al., 2015) and separately for SIF product contain one day of data per file (Frankenberg, 2015).  For $X_{CO2}$ a
bias correction is applied, warn levels are assigned, with all converged soundings included in the file.

### 4.3.1  L2 $X_{CO2}$ results

The $X_{CO2}$ data record from OCO-2 now extends more than 18 months, and Figures 3, 4, and 5 show maps of these $X_{CO2}$
measurements. These maps illustrate averages over month long periods, so there are nadir and glint data in each panel. The data
included in these maps and all that follow have been screened and have had the bias correction applied (v7rB Lite file data with
the 0/1 data quality flag applied, see Mandrake et al., 2015). These two processes will be discussed in more detail in Sections
4.3.4 and 4.3.6. As expected, these maps show the large annual changes in $X_{CO2}$. $CO_2$ builds up over the Northern Hemisphere
during winter, and then is rapidly removed from the atmosphere as spring arrives and the terrestrial ecosystem activity increases
rapidly. This is most apparent in the month of June, when the decrease of $X_{CO2}$ over northern Asia is order 10 ppm. The overall
gradients of a few ppm from north to south are apparent in the data, as well as the secular increase in $CO_2$ from October 2014 to
March 2016. Other features are apparent in the data maps, such as the higher $CO_2$ concentrations over the Eastern US and China
between October and December (see Figures 3 and 5), when the overall global $X_{CO2}$ gradient is small. Enhanced $X_{CO2}$ coincident
with biomass burning in the Amazon, central Africa, and Indonesian is also obvious in these figures.
The latitudinal coverage of the v7r dataset is also apparent from these maps. Data selection for processing through L2 relies on
screening from the preprocessor results, as well as limitations on geographical extent. Analysis of the preprocessor data (Taylor
et al., 2016) show that a large fraction of these higher latitude data are marked as cloudy, which is in agreement with the MODIS
cloud fields. The current data selection does not select data south of 65 degrees in latitude, as experience with ACOS data
showed that retrievals over ice failed routinely. We intend to retrieve the small number of cloud-free scenes over bare ground at
these latitudes in the next version of the retrieval. Due to clouds, solar illumination and geometry, any given month has data that
spans about 100 degrees in latitude, but the coverage band shifts north and south with the seasons.

### 4.3.2  Signal to noise ratios

The OCO-2 instrument was designed to provide adequate continuum signal to noise (SNR) to achieve 0.3% precision for $X_{CO2}$
measurements. The SNR design requirements were 290, 270, and 190 at nominal radiance levels in the A-band, weak and strong
$CO_2$ band, respectively. The in-flight performance has met or exceeded all expectations, with SNR values as provided in the data
product (radiance mean value in the continuum divided by the radiance noise value in the continuum) typically between 250 and
450 for the A-band, 400 to 800 for the WCO2 band, and 200 to 500 for the SCO2 band.  Figure 6 illustrates just one month of
SNR levels, as no large seasonal dependence is observed. There are spatial patterns, with high SNR values over the bright deserts
and in cloudy regions. The lowest SNR values are over oceans, especially when observed at higher solar zenith angles,
particularly for the A-band.

### 4.3.3  $\chi^2$ goodness of fit parameter




$$\chi_i^2 = \frac{\frac{1}{n}\Sigma(y - F(x))^2}{\frac{1}{n}\Sigma\varepsilon^2} \qquad (4\text{-}1)$$

The reduced $\chi^2$ goodness of fit parameter is a convenient measure of the magnitude of the spectral residuals relative to the measurement error. The equation for per band ($\chi_i^2$) is given in Equation (4-1) where where $i$ is the band index (1..3), $y$ is the measured radiance spectrum, $\varepsilon$ is the error on the measured radiance spectrum, and $F(x)$ is the forward model with the state vector $x$ (Crisp et al., 2014; O'Dell et al., 2012, 2016). As discussed in Crisp et al., 2014, the persistent spectral residuals caused by limitations in the spectroscopic input data and instrumental effects are removed by fitting to empirically derived spectral vectors. This approach systematically reduces $\chi^2$ and also reduces the dependence of $\chi^2$ on the signal to noise ratio.

For OCO-2, we have seen that there is little seasonal dependence, but there are clear spatial patterns, as illustrated in Figure 7. In the A-band, prominent features occur in the region of the South Atlantic Anomaly (SAA) (Crisp et al., 2016a). The effects of this region of a high density of high energy particles are seen as radiance spikes in the A-band measurements. We attempt to screen out the effects, but the fitting is still poor in this region. For the weak and strong $CO_2$ bands, the bright desert of the Sahara results in larger chi-square values, and mountainous regions impact the strong $CO_2$ fits.

### 4.3.4 Warn levels

The data presented in this paper have data quality screening applied. For the OCO-2 dataset, we have developed warn levels (Mandrake et al., 2013 and Mandrake et al., in prep 2016). The concept behind the warn levels is that the data are ordered by quality as defined by a number of data variance metrics, allowing the user to make decisions concerning the trade off between data volume and data quality. This is a more flexible approach then the traditional good or bad quality assignment, and reflects the fact that data quality is a continuum, not a binary quantity, and should be indicated as such. The OCO-2 warn levels range from 0 to 19, with 0 indicating the highest quality and 19 considered the lowest quality. More details of the process used to develop warn levels are reported in Mandrake et al. 2013, 2016 (in prep) as well as the OCO-2 Lite file documentation [Mandrake et al., 2015]. Our recommendation is that users should not use data above a warn level of 15 for all land data, nor above 18 for water glint. This removes approximately 25% of the land data and 10% of the water glint data.

For the v7r data, outliers were screened with a set of additional flags, related to the cloud preprocessors, aerosol optical depths, surface characteristics, etc. The detailed flagging parameters and thresholds are provided in the Lite file user's guide. The warn level thresholds and outlier screening are combined in the 0/1 flag that is included in the Lite file, to be compatible with the European Greenhouse Gas Climate Change Initiative (GHG-CCI) data product specifications (Buchwitz et al., 2015). We have used this screening for the maps shown in this paper, but strongly encourage users to carefully evaluate the warn levels that are appropriate for their science analysis.

### 4.3.5 Data density after quality screening

The data density after quality screening for a few select months is illustrated in Figure 8. The monthly total data density ranges from 1.3 million to 2.4 million soundings per month selected by the xco2_quality_flag in the Lite file for periods without decontamination cycles, influenced by the mixture of nadir and glint measurements, as well as clouds and season. For individual



2 degree by 2 degree regions, the number of soundings in a month range from a few to over a thousand. There is a roughly
inverse relationship, so for example, on a monthly basis, about 100 of the 2 degree by 2 degree cells have 100 soundings, and 10
have 1000 soundings. The preprocessors, as described in Taylor et al. (2016), limit the data that is put through L2 processing, and
then processing failures and data screening further trim the dataset. Nevertheless, there is a large volume of high quality data
available from OCO-2. The highest densities of data are over desert areas, although mid-latitude data density is high during some
seasons. As reported in Taylor et al. (2016) the prescreening and resulting data density is consistent with MODIS cloud statistics.
The cloudy region of the inter-tropical convergence zone (ITCZ) has lower data density, as does northern South America. This
region is impacted by clouds as well as the SAA, where cosmic ray events impact OCO-2 measurements. For the v7/v7r data, the
preprocessors do not account for the SAA impacts, and thus a significant fraction of data are screened out. In the next version,
the preprocessors will have SAA treatment integrated, and we expect that the data yield will increase in this region.
### 4.3.6  Bias correction
The bias correction described in O'Dell et al. (2016) and the OCO-2 documentation (Mandrake et al., 2015) was applied to the
$X_{CO2}$ data shown in Figures 3, 4, and 5. The monthly mean bias corrections for 3 sample months are shown in Figure 9. The bias
correction seeks to remove systematic footprint-to-footprint differences, mode-to-mode differences (for example systematic
differences between land glint and land nadir measurements), and systematic differences that appear to be correlated to other
retrieval variables. Two predictive variables are currently used in the bias correction for land retrievals, and three are used for
ocean retrievals. In addition, the bias correction process puts the OCO-2 data on the same scale as the Total Carbon Column
Observing Network (TCCON) ground-based measurements, which are tied to the WMO scale for carbon dioxide (Wunch et al.,
2016, 2010, 2011). The OCO-2 mission development included a validation plan which recognized the need for the TCCON
network and a special data collection mode to gather adequate validation data. A detailed discussion of the ground-based data and
the OCO-2 data that are collected in target mode at these locations can be found in Wunch et al., 2016. Details of the derivation
of the bias correction and its relationship to other variables can be found in O'Dell et al. (2016). The monthly distribution of the
bias correction values are well described by Gaussian distributions. Overall, for the water glint observations on monthly scales,
the mean of the distribution is 0.0 to 0.4 ppm, with a standard deviation of about 0.55ppm. For land glint observations, the mean
is larger, 0.9 to 1.1 ppm, and the standard deviation is typically 1.2 ppm. The land nadir distribution has a similar standard
deviation, about 1.2ppm, with a mean of 1.3 to 1.8 ppm.  The patterns strongly follow latitudinal gradients, likely driven by
viewing geometry with aerosol and cloud scattering becoming more important as the instrument views through longer paths of
the atmosphere. The bias correction is described in more detail in O'Dell et al. (2016).
### 4.3.7  Uncertainty on $X_{CO2}$ product
The OCO-2 data products include an estimate of the uncertainty on the $X_{CO2}$ data. As discussed by Connor et al. (2008, 2016),
this estimate is a lower bound, as it includes error related to the noise on the radiance measurement, the smoothing error, and
interference error. Propagation of systematic errors in input terms for the forward model to the $X_{CO2}$ estimate is not considered in
the error estimate reported in the v7/v7r L2 products. Figure 10 is a set of maps of the average $X_{CO2}$ uncertainty from the data
product for a six-month period. This shows that the estimated uncertainty is generally smaller over water than the land surface
and that the uncertainty is larger at the extreme latitudes, where interference errors grow. Worden et al. (2016) have made a
careful assessment of the OCO-2 uncertainty estimates, by evaluating the standard deviation of the difference from the mean
$X_{CO2}$ for collections of soundings within 100km in latitude. They compare this to the expected standard deviation due to noise.



This research showed that while linearly correlated, the $X_{CO2}$ calculated measurements error in the data product appears to
underestimate the empirically-derived $X_{CO2}$ measurement error by a factor of approximately 2, with a larger underestimate for
land data and a smaller underestimate for water glint measurements.
In the optimal estimation retrieval, algorithm input choices such as the *a priori* mean state vector ($\mathbf{x_a}$) and *a priori* covariance
($\mathbf{S_a}$), or constraint, can impact the variability in the retrieval error in $X_{CO2}$. The *a posteriori* covariance matrix ($\hat{\mathbf{S}}$) is also an
important output of the L2 retrieval process, as it is critical for the data assimilation process used to determine $CO_2$ fluxes. The
OCO-2 project is in the midst of an evaluation of this quantity and the accuracy of the algorithm's reported uncertainty as a
measure of the error variability, through the use of large-scale simulations. By running simplified retrievals over large ensembles
of input variables (priors, constraints, and other parameters), one can assess the characteristics of the retrieval bias and variance
and evaluate what is reported in the data product (Hobbs et al., 2016). The choice of prior becomes particularly impactful for
moderate to large aerosol optical depths (0.1 or more).
There are many other variables that are co-retrieved with the $X_{CO2}$, including surface pressure, aerosol optical depth, surface
albedo, water profile scaling factor, and an offset of the temperature profile. The aerosol optical depths are being compared
against independent measurements, such as AERONET optical depths while an analysis of the retrieved water vapor profiles
against SuomiNet and the Advanced Microwave Scanning Radiometer-2 (AMSR-2) is also being conducted (Nelson and O'Dell,
2016). As discussed in detail in O'Dell et al. (2016), many of these parameters will compensate for one another in the retrieval
algorithm, so must be considered 'effective quantities' (e.g. 'effective albedo' and 'effective optical depth') as they are the values
that minimize the fit in an optimal estimation scheme, but they are at times not directly related to the physical quantity (Kulawik
et al., 2006, Eldering et al., 2008). The performance and relationships of these parameters are discussed at length in O'Dell et al.,

20  2016.

**4.4   Solar induced fluorescence**
Using GOSAT and GOME-2 spectra, Frankenberg et al. (2011, 2012, 2014, 2015; Joiner at al. 2011) demonstrated that using the
observed Fraunhofer line fractional depths, solar-induced fluorescence of chlorophyll can be quantified. Frankenberg et al.
(2014) performed a pre-flight assessment of the fluorescence measurement performance of OCO-2. This measurement approach
is being applied to the OCO-2 data, motivated in part because neglect of this phenomenon results in errors in surface pressure and
aerosol optical depth, which propagate into a small bias in the $X_{CO2}$ retrieval (Frankenberg et al., 2012).
The IDP preprocessor performs the SIF retrieval, along with single band retrievals of the water and $CO_2$ columns that are used
for cloud screening purposes. As described in Frankenberg et al. (2014) the SIF retrieval is impacted less strongly by clouds than
the $X_{CO2}$ retrieval, so useful data is collected over a much larger number of soundings. However, high single-measurement
precision errors warrant aggregation in space and/or time for scientific use. The SIF product is derived at two wavelengths,
757nm and 771nm, and it is recommended that the user examine both fields independently, as this first dataset (v7r) may have
different errors in each product.
Figures 11 illustrates a year of SIF retrievals, where data has been averaged across seasons. These show expected features, such
as the high SIF values in the regions of intense agriculture during early summer, and the low SIF in the Northern Hemisphere
during its winter. The SIF signal in the tropics has some seasonality to it, but is always larger than 0.5 W m$^2$ μm$^{-1}$ sr$^{-1}$.



Campaigns are underway to compare OCO-2 measurements to data at flux towers and to underfly the OCO-2 measurements with
an aircraft-mounted grating spectrometers. Details of these inter-comparisons are in Sun et al. (in prep, 2016). The objective of
those studies is to quantify the relationship of OCO-2 derived SIF with independent measurements.
## 5   Gradients and trends in observed $X_{CO2}$
### 5.1   Growth rate of $X_{CO2}$
The dense, global dataset from OCO-2 can be used to assess the annual growth rate of $X_{CO2}$. Figure 12 shows the annual zonal
growth rates derived from OCO-2 for 5 different 12 month periods. The growth rate as determined from the NOAA ESRL station
at Mauna Loa is shown for comparison. The growth rates are generally between 2.5 and 3 ppm per 12 months from 2014 to 2015,
which includes the largest growth rate ever recorded at the Mauna Loa Observatory. The figure also illustrates the longitudinal
standard deviation of the OCO-2 data for each latitude band. Note that the Mauna Loa Observatory is a background site, whereas
the OCO-2 measurements span both background sites and populated regions, and this variability drives the standard deviation.
The relative sampling of regions of emissions and uptake differs in time with OCO-2, which will result in a different 12-month
growth rate than that derived from the NOAA ESRL station.
### 5.2   Seasonal cycle of $X_{CO2}$ near Hawaii
A time series of weekly average $X_{CO2}$ from OCO-2 for a region around Hawaii is shown in Figure 13.  For this analysis, we have
selected glint water data only, and gathered data from a region that span from 175 W to 130W in longitude, and from 15N to 25N
in latitude. The time series clearly shows weekly and monthly changes as observed by OCO-2. The ground-based measurements
collected at Mauna Loa Observatory are overplotted, although OCO-2 data are not directly comparable to the NOAA ESRL and
other ground-based measurements, because OCO-2 senses the total column of $X_{CO2}$ rather than surface concentrations. If the
vertical gradient of $CO_2$ is small, we expect similar values for the two measurements, whereas the surface measurements will be
larger then the OCO-2 $X_{CO2}$ measurements if there are $CO_2$ enhancements in the lower atmosphere. Both datasets show little
growth between January and February 2015, and in early 2016. The surface measurements and $X_{CO2}$ are most similar in August
and September, and the $X_{CO2}$ further north is nearly the same as Mauna Loa surface $CO_2$ concentrations in August through
October. The differences between the total column and surface measurements are greatest in the early spring, as the build-up of
$CO_2$ in the Northern Hemisphere is present in the near surface layers, but has not propagated vertically to the $X_{CO2}$ signal. The
standard deviation of the weekly-averaged data range from 0.5 to 0.8 ppm, with 2000 to 20,000 measurements averaged per week
for the OCO-2 data.
### 5.3   Latitudinal gradient
The OCO-2 record adds additional detail to our understanding of the latitudinal gradients of $X_{CO2}$. Figure 14 shows the zonal
means of quality flagged OCO-2 $X_{CO2}$ data. As expected, there is a complete reversal of the latitudinal gradient between
Northern Hemisphere spring and Northern Hemisphere summer.  We see the contrast of the March-April 2015 gradient, where
the Northern Hemisphere has $X_{CO2}$ concentrations 4 to 7 ppm larger than the Southern Hemisphere, whereas the reverse is seen





in July - August. The southern hemisphere gradient is similar from September through April. The seasonal change of latitudinal
coverage is also apparent in this plot, driven by both solar geometry and clouds, as discussed earlier. Wunch et al. (2016) shows
comparisons of the latitudinal gradients as observed by OCO-2 and TCCON.

## 5.4   Assessment of overall data quality

The OCO-2 mission has been successful in collecting over a million measurements of radiance spectra in the A-band, weak and
strong $CO_2$ bands each day. After screening for clouds, and applying post retrieval quality flags, OCO-2 typically delivers
100,000 global measurements of $CO_2$ per day. Detailed comparisons have been made against the Total Column Carbon
Observing Network, and the OCO-2 measurements agree within 1 ppm in most cases (see Wunch et al., 2016).
There are regions of the world that have consistent high data yields, such as desert regions and the oceans to the north and south
of the cloudy ITCZ. Regions of persistently low data yield include the region over South America that is impacted by the South
Atlantic Anomaly, ocean regions of the ITCZ, and regions where the solar zenith angles are large (especially northern latitudes in
NH winter, and southern latitudes during SH winter).
The dataset is consistent in time, showing stability in diagnostic parameters such as the measurement SNR and retrieval $\chi^2$ as
well as the overall data density. Not surprisingly, there are some data features that are inconsistent with the validation dataset,
and different from model predictions. The largest feature is a high bias in $X_{CO_2}$ over water for southern latitudes during the
Southern Hemisphere winter. This issue is apparent in the TCCON comparisons for Wollongong shown in Wunch et al. (2016),
and in the comparison to models presented in O'Dell et al. (2016). This bias has been extensively examined by the OCO-2 teams,
who have considered viewing geometry, polarization effects, interferents such as aerosols, surface models, and instrument
performance. The analysis has not yet yielded insights into the root cause, although in early testing, there are indications that the
lack of stratospheric aerosols in the current version of the retrieval algorithm can significantly increase bias.
The v7/v7r data version discussed here is the current operational data product. In the future, a v8/v8r data product will be
produced that addresses calibration issues as described in Crisp et al. (2016ab), as well as retrieval algorithm improvements
described in O'Dell et al. (2016) such as the land surface treatment and others that are not yet fully tested. Future changes to the
retrieval algorithm will focus on improving the parameterization of the patterns of bias for correction, if not direct reduction of
the bias.

## 6   Conclusions

The OCO-2 mission has been successful in collecting a dense, global set of high-spectral resolution measurement that are used to
estimate the column-averaged atmospheric $CO_2$ dry air mole fraction, $X_{CO_2}$. The first 18 months of the missions have provided
1.3 to 2.4 million $X_{CO_2}$ measurements per month after screening for data quality. As described in Wunch et al. (2016), the data
have median difference of less than 0.5 ppm with the primary ground-based validation network, and RMS differences typically
below 1.5 ppm. Large-scale features, such as the drawdown of $CO_2$ in the Northern Hemisphere spring and the increase of $CO_2$
over Northern Hemisphere winter are obvious in the data. By meeting the mission goals for accuracy, resolution, and coverage,
the OCO-2 mission has provided a dataset that can now be used to assess regional-scale sources (emitters) and sinks (absorbers)
around the globe.





**7   Data Availability**
All of the OCO-2 data products are publically available through the NASA Goddard Earth Science Data and Information
Services Center (GES DISC) for distribution and archiving (http://disc.sci.gsfc.nasa.gov/OCO-2).
**Acknowledgements**
Part of this work was conducted at the Jet Propulsion Laboratory, California Institute of Technology under contract with the
National Aeronautics and Space Administration (NASA) for the Orbiting Carbon Observatory-2 Project. Work at Colorado State
University and The Geology and Planetary Sciences Department at the California Institute of Technology were supported by
subcontracts from the OCO-2 Project. The NOAA ESRL ground-based weekly averaged measurements from Mauna Loa
Observatory were obtained from www.esrl.noaa.gov/gmd/ccgg/trends/.





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



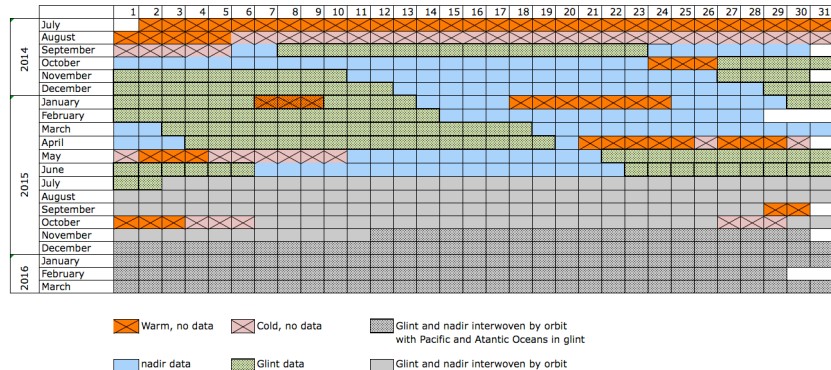

5 Figure 1. OCO-2 Data calendar with observation modes and data outages.

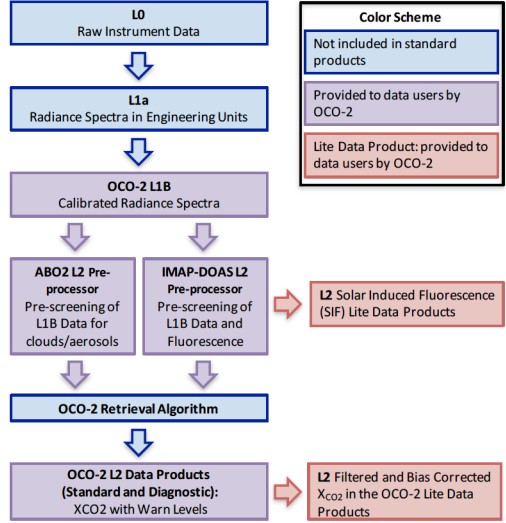

8 Figure 2. OCO-2 Data processing flow.




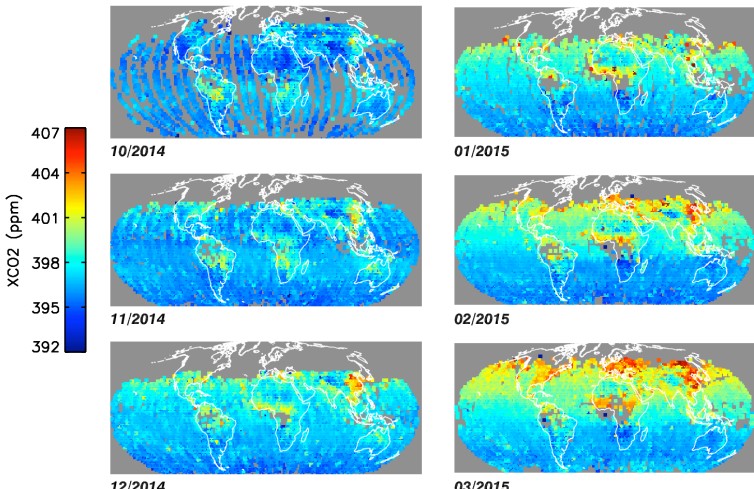

Figure 3. Maps of total column dry air ratio of $CO_2$ ($X_{CO2}$) from OCO-2 from October 2014 through March 2015. Data has been
bias corrected and screened using the data quality flag in the Lite file, and averaged in 2 degree by 2 degree bins.

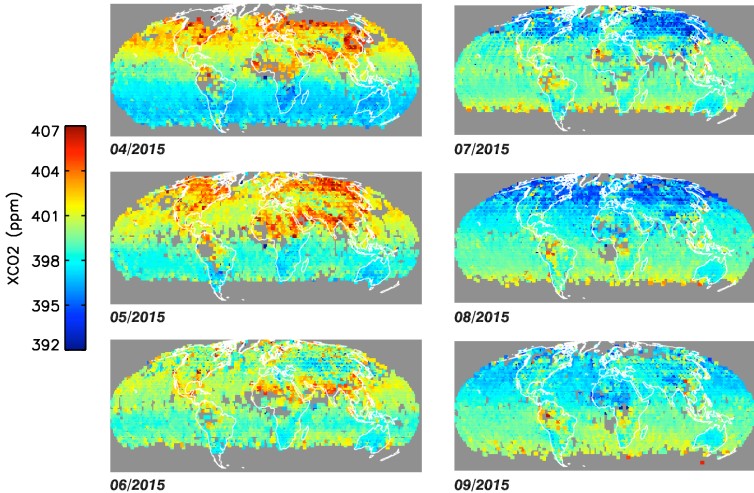

Figure 4. Maps of $X_{CO2}$ from OCO-2 from April 2015 through September 2015, bias corrected and selected with data quality flag
and averaged on 2 degree by 2 degree grid.

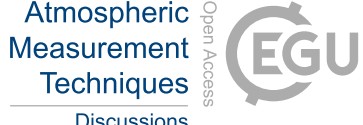

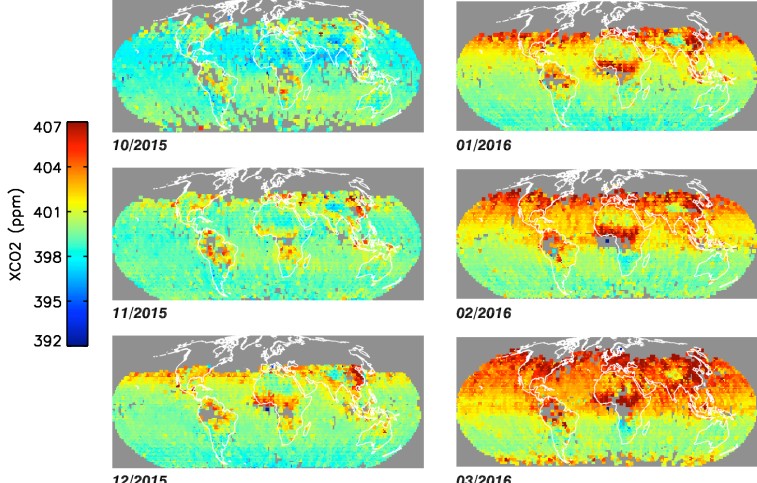

3    Figure 5. Maps of $X_{CO_2}$ from OCO-2 from October 2015 through March 2016, bias corrected and selected with data quality flag

4    and averaged on 2 degree by 2 degree grid.





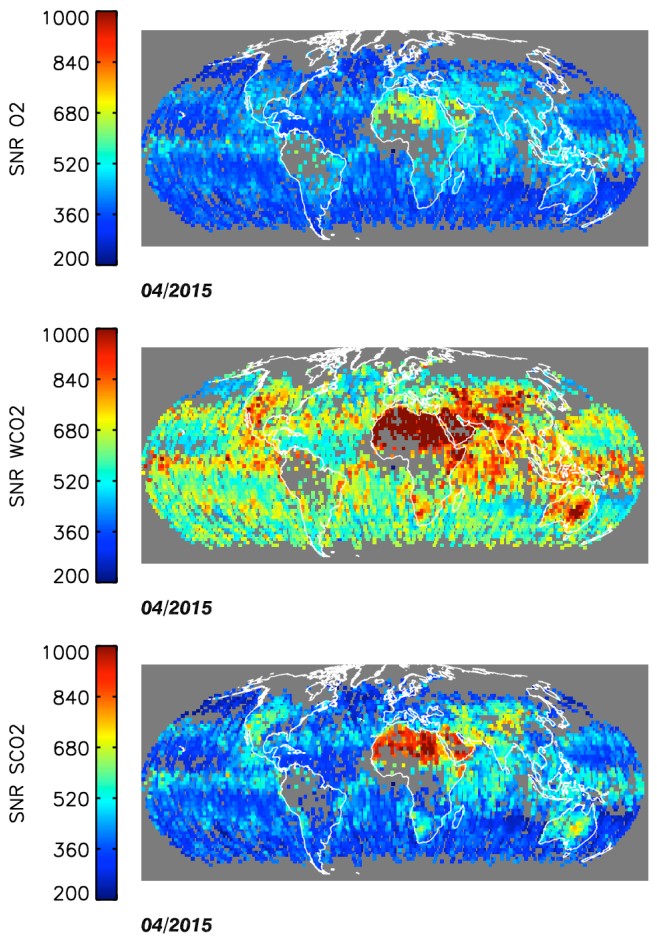

2  Figure 6. Maps of the continuum signal to noise ratio for the three bands of the OCO-2 instrument in April 2015. Statistics are

3  provided for 2 degree by 2 degree bins for data selected with the data quality flag.





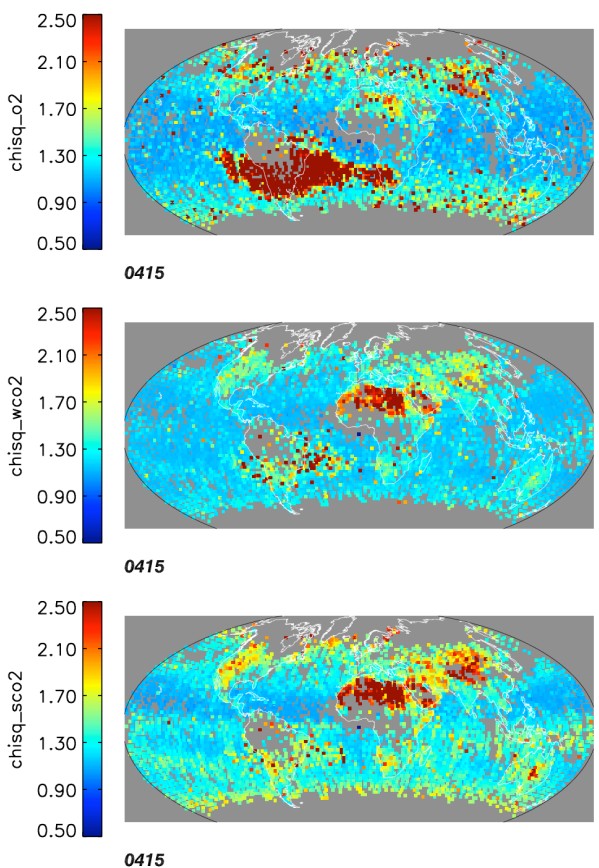

2    Figure 7. Maps of the fitting parameter $\chi^2$ three bands of the OCO-2 instrument in April 2015. Statistics are provided for 2 degree

3    by 2 degree bins for data selected with the data quality flag.





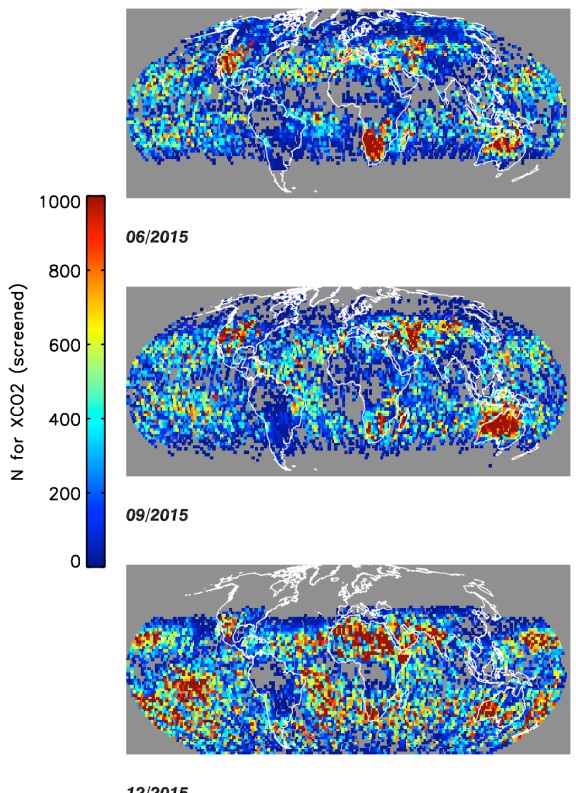

2    Figure 8. Maps of the number of soundings passing quality flagging for a selection of months. Statistics are provided for 2 degree

3    by 2 degree bins for data selected with the data quality flag.





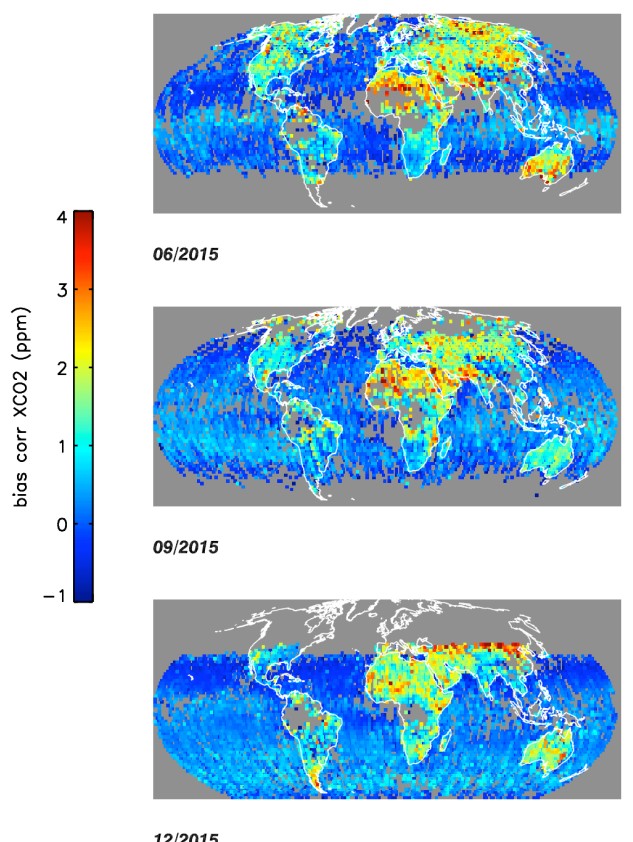

2    Figure 9. Maps of the bias correction applied to the $X_{CO2}$ data. Statistics are provided for 2 degree by 2 degree bins for data

3    selected with the data quality flag.





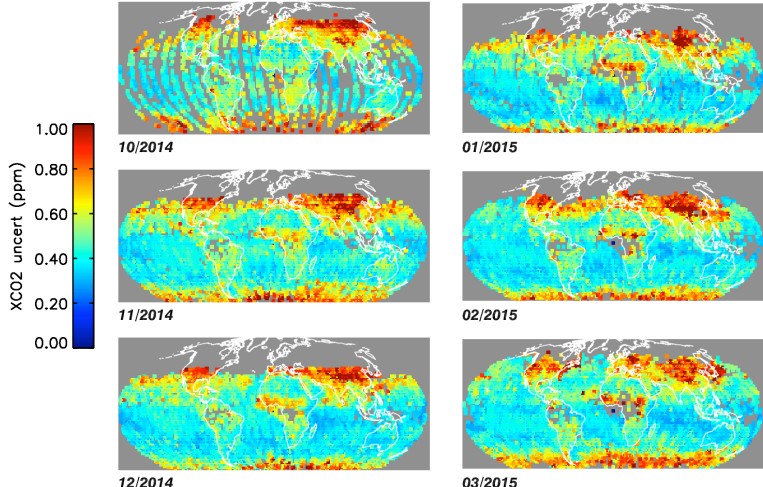

2     Figure 10. Maps of the average $X_{CO2}$ uncertainty in the OCO-2 data product. Statistics are provided for 2 degree by 2 degree bins

3     for data selected with the data quality flag.




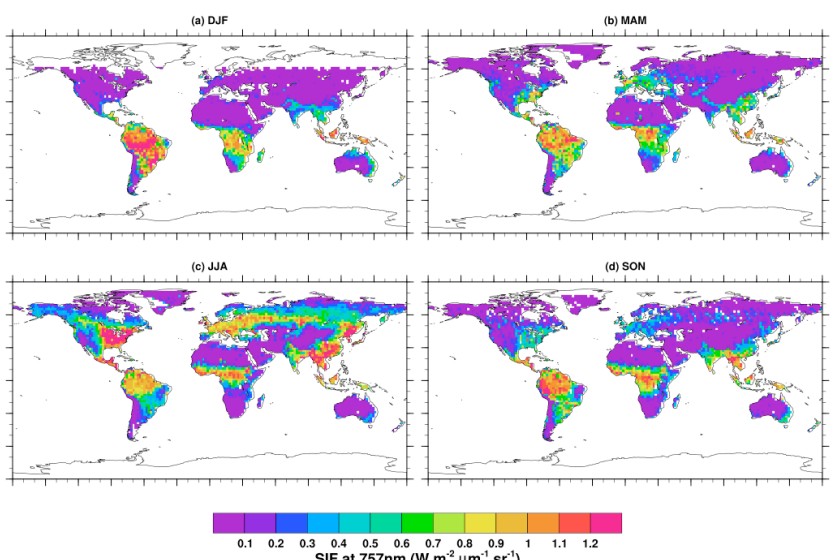

2 Figure 11 – OCO-2 Solar Induced Fluorescence (SIF) product averaged on 2 degree by 2 degree grid for 3 month periods

3 (December 2014 through November 2015).

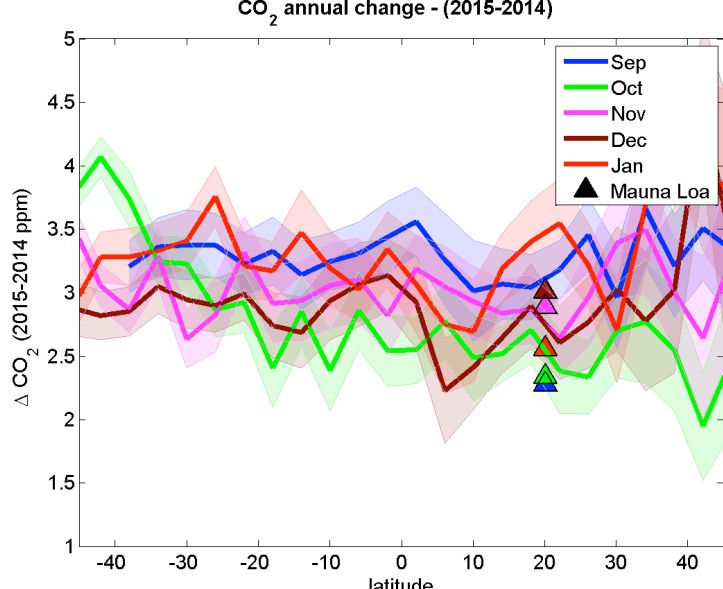

2    Figure 12. Annual change of $X_{CO2}$ zonal means from OCO-2 observations (lines) and from in situ measurements at Mauna Loa,

3    Hawaii (triangles), plotted in different colors for the different months of measurements. The shaded areas represent the standard

4    deviations.




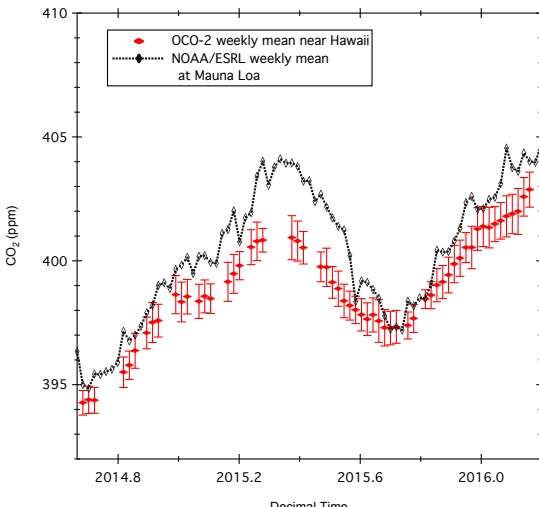

3    Figure 13 – Time series of weekly average OCO-2 $X_{CO2}$ measurements near Hawaii. Glint water measurements selected with the

4    data quality flag from the Lite files. NOAA ESRL Mauna Loa Observatory weekly average $CO_2$ concentrations are overplotted

5    in a black dashed line.





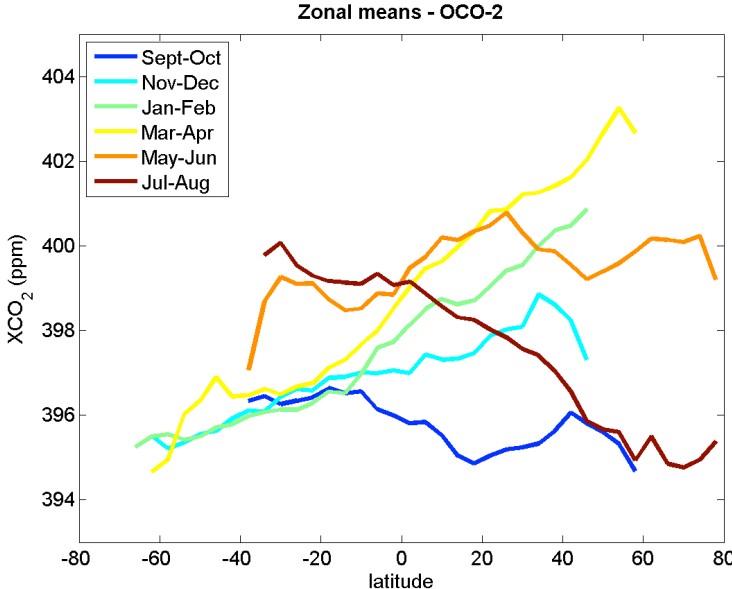

2  Figure 14: Latitudinal gradient for 2 month averages of OCO-2 $X_{CO2}$ data between September 2014 and August 2015. All modes

3  are averaged together in these figures.

