# Peer review of "The Orbiting Carbon Observatory-2: First 18 months of Science Data Products"

_Atmospheric Measurement Techniques, 2016_

## Referee Comment (RC1) · Anonymous Referee #1 · 9 Oct 2016

Manuscript "The Orbiting Carbon Observatory-2: First 18 months of Science Data Products", Atmos. Meas. Tech. Discuss., doi:10.5194/amt-2016-247, 2016, from Eldering et al., presents an overview about the OCO-2 Level 2 data products focusing on the column-averaged CO2, i.e., XCO2, data product. The manuscript is well written contains important overview information, which is likely very useful for users of the OCO-2 data products. The topic is appropriate for Atmos. Meas. Tech. and I recommend publication after the comments listed below have been carefully considered by the authors when generating the revised version of the manuscript.

Specific comments:

Abstract: I recommend to add information on which version of the data products is presented (v7/v7r, Lite files) and which time period (September 2014 – January 2016

?). I find this more important than launch date and the data when OCO-2 joined the A train (it is sufficient to mention the latter in the Introduction).

Page 3, line 21 following: I recommend adding information on where (approx.) the shape is nearly rectangular (high latitudes?) and where it is very narrow (tropics?).

Page 7, line 17-18: Sentence "Enhanced XCO2 coincident with biomass burning in the Amazon, central Africa, and Indonesian is also obvious in these figures.". Sorry, but this statement is not supported by the figures shown, which only display XCO2 but no (independent) information on biomass burning. Please provide more evidence to support that statement or revise the statement.

Page 11, line 13: Statement "and this variability drives the standard deviation". The standard deviation could also be driven by biases. How can you be sure that this is not the case? Please provide more evidence to support that statement.

Page 11, Section 5.2: The difference of the two curves shown in Fig. 13 is 2-3 ppm which is a large difference for a CO2 seasonal cycle. Please mention this and comment on this.

Page 11, line 32: Sentence "The OCO-2 record adds additional detail to our understanding of the latitudinal gradients of XCO2.". Really? Please list explicitly what the new knowledge is or revise this statement. Figure 14 shows a large difference between southern hemisphere XCO2 between the two time periods Sept-April (blue, yellow) and May-Aug (orange, red). Is this assumed to be a real feature?

Page 12, line 33: Please add more info on the 1.5 ppm. Does this number correspond to bias-corrected and quality filtered single observations as contained in the v7/v7r Lite files?

Page 28, Figure 12: Why only Mauna Loa for comparison? Would be interesting to also see comparison with NOAA data for other latitude bands (e.g., to see if the latitudinal dependence in October is consistent with NOAA or not). Please check figure title (I

guess Jan means "Jan 2016 – Jan 2015" although the title suggests 2015 - 2014).

Typos etc.:

Page 1, lines 10-13: The curly braces "{}" need to be removed, I think.

Page 3, line 5: remove empty space after "4.3.1)".

Page 5, line 36: Check sentence (". . . here, data . . .")

Page 29, Figure 13: Please improve x-axis numbering so that it is easier to see the beginning of each year.

Pages 14 – 18: References: Please check all references carefully: Needs some harmonization, e.g., with respect to authors list: Currently it is (i) a mix of listing only one author, or several or all (sometimes even with ". . ."), and (ii) author's given name is typically abbreviated but not always, etc. Furthermore mix of "in preparation" and "(in prep)".

---

## Short Comment (SC1) · 13 Nov 2016

We appreciate the comments of the reviewer.

We have read and digested all of the reviewer's comments, and are preparing an updated version of the manuscript that addresses these. In some cases, we will work with our co-authors to generate updated figures.

Regarding the question related to growth rates. The NOAA-ESRL official website provides only growth rates at Mauna Loa and as a global average, this we have only included Mauna Loa in this papers (see http://www.esrl.noaa.gov/gmd/ccgg/trends/gr.html).

More detailed discussion of the comparison of global models and the OCO-2 data will

be reported by other scientists from the global modeling community.

We will update our manuscript with responses to all of the reviewer's comments and upload this week.

thanks for your time and interest,

Annmarie Eldering

---

## Referee Comment (RC2) · Anonymous Referee #2 · 21 Nov 2016

The manuscript of Eldering et al. provides a nice overview over the data products from the NASA OCO-2 satellite missions and it gives an illustration of some first results. The publications will very informative and beneficial for anybody interested in the OCO-2 data. The article is rather a review article than a strict science publication and is summarizing results from a number of companion papers (many of those are either in review or in preparation). As a consequence, the manuscript is limited in terms of novel science findings but it will be nevertheless a very valuable reference for the science community. One downside is that the manuscript discusses the data release v7 which will be superseded shortly with the release of the v8.

I have some comments/suggestions, mainly related to section 5 that the authors should address. There is also a number of minor corrections that need to be applied.

[Figure]

What is the reason of showing the growth rate per latitude as figure 12 seems to show very little latitudinal difference. Can you please discuss what you would expect to find and what you do see from OCO-2. What can we learn from the comparison of column data from OCO-2 to Manu Loa in-situ data? Both don't agree very well (eg for Sep the difference is almost 1 ppm) so I don't understand if you try to say that OCO-2 data shows a reasonable growth rate or not. At least, you could select only OCO-2 data around Mauna Loa. Clearly, much more useful would be a comparison to the growth rate from OCO-2 data a TCCON station.

Very similar questions apply to figure 13. This is an 'apples to oranges' comparisons. I think it would be useful to add column data from a model constraint with in-situ data such as CarbonTracker or CAMS so that proper column to column comparisons are possible.

Also, figure 14 does not add much value as no context is given. Adding data from a model to the figure (see above) would allow to gauge if the OCO-2 is roughly consistent with expectations.

Minor comments: p.1 : Correct the assignment of affiliations of authors

p.2, l. 9: For mass balance. . . -> For mass balance reasons. . .

p1. L. 34 measure atmospheric $CO_2$ with -> measure atmospheric $CO_2$ columns with

p.5 l. 35: error of 4% will impart an $XCO_2$ error of 0.22 ppm, 0.12 ppm, and 0.4 ppm -> Connor et al., AMT, 2016 seems to suggest a much larger error in $CO_2$ as a result of 4% radiometric uncertainty. How did you calculate this $CO_2$ error?

p.6 (optical depths less than $\sim$0.35) -> does this refer to the retrieved optical depth which must not agree with the true optical depth?

p.7 SNR design requirements were 290, 270, and 190 at nominal radiance levels -> please give the nominal radiance levels

p.9 eq. 4-1: y, x, F, e are vectors and should be given in bold or use index to show that this is a sum over elements of the vector. Vector should be given in bold in the following paragraph as well. Also, define n.

p. 11 l. 19: selected glint water data only -> selected glint data over water only

p. 11, l.19 175 W to 130W in longitude, and from 15N to 25N -> 175ˆo W to 130ˆo W in longitude, and from 15ˆo N to 25ˆo N

p. 17, l.5: retrieval -> retrieval

p. 17, l. 5-8: There are 2 references labelled O'Dell et al., 2016 Can you confirm that Fig. 6-9 show the same coverage as Fig.. 3-5. It looks different but this might simply due to the different figure size.

---

## Author Comment (AC1) · 23 Dec 2016

We appreciate the time that referee #1 invested in our paper. This comment contains detailed responses to their review. A manuscript with tracked changes will be attached in a later comment.

Response: We appreciate the detailed review from anonymous reviews #1, and the reviewer clearly understands our intentions with this paper – to document the mission and current status. We have made a number of edits to the paper based on the comments, which are detailed below.

Comments from anonymous reviewer #1 and author responses

Abstract: I recommend to add information on which version of the data products is

presented (v7/v7r, Lite files) and which time period (September 2014 – January 2016 ?). I find this more important than launch date and the data when OCO-2 joined the A train (it is sufficient to mention the latter in the Introduction).

Response: we have made this revision to the abstract

Page 3, line 21 following: I recommend adding information on where (approx.) the shape is nearly rectangular (high latitudes?) and where it is very narrow (tropics?).

Response: we have modified the description in section 2 to make this more clear.

Page 7, line 17-18: Sentence "Enhanced XCO2 coincident with biomass burning in the Amazon, central Africa, and Indonesian is also obvious in these figures.". Sorry, but this statement is not supported by the figures shown, which only display XCO2 but no (independent) information on biomass burning. Please provide more evidence to support that statement or revise the statement.

Response: We have added a citation of Van Der Werf (2010) which clearly shows the location and seasonality of the biomass burning that we refer to.

Page 11, line 13: Statement "and this variability drives the standard deviation". The standard deviation could also be driven by biases. How can you be sure that this is not the case? Please provide more evidence to support that statement.

Response: This was a useful comment – we looked in more detail at the variability of the retrievals, and added a statement clarifying that this could be due to the variability of sources, but land retrievals are more variable then the ocean glint, and that is another possible explanation of the standard deviation.

Page 11, Section 5.2: The difference of the two curves shown in Fig. 13 is 2-3 ppm which is a large difference for a CO2 seasonal cycle. Please mention this and comment on this.

Response: Both reviewers had comments about this figure, focused on the 'apples to

orange' comparison of total columns and surface measurements. It would be significant additional scope to discuss the vertical distributions, averaging kernels, and how to properly compare the surface data to the total columns. Therefore, we have decided to limit this discussion to just OCO-2 data, showing a weekly timeseries and pointing out the standard deviation of the OCO-2 data relative to the changes in time. We have also improved the time axis on the graph.

Page 11, line 32: Sentence "The OCO-2 record adds additional detail to our understanding of the latitudinal gradients of XCO2.". Really? Please list explicitly what the new knowledge is or revise this statement. Figure 14 shows a large difference between southern hemisphere XCO2 between the two time periods Sept-April (blue, yellow) and May-Aug (orange, red). Is this assumed to be a real feature?

Response: Based on your comments and the comments of another reviewer, we have decided to eliminate Figure 14 from the paper. You are correct in that more detailed are needed to explain what we have learned and what questions remain about the observed gradients, and we will leave that to later publications.

Page 12, line 33: Please add more info on the 1.5 ppm. Does this number correspond to bias-corrected and quality filtered single observations as contained in the v7/v7r Lite files?

Response: We have added a sentence stating explicitly what data is used in the Wunch et al analysis, where that figure is reported.

Page 28, Figure 12: Why only Mauna Loa for comparison? Would be interesting to also see comparison with NOAA data for other latitude bands (e.g., to see if the latitudinal dependence in October is consistent with NOAA or not). Please check figure title (I guess Jan means "Jan 2016 – Jan 2015" although the title suggests 2015 - 2014).

Response: We have updated the figure title and figure caption - you are correct, Jan is a 2016-2015 value, and the others are 2015-2014. NOAA ESRL only reports growth

rates at Mauna Loa, and thus we have included only that data. There are other papers that have examined the growth rate from similar remote sensing data, and we now cite those in this paper.

Typos etc.:

Page 1, lines 10-13: The curly braces "{}" need to be removed, I think. Page 3, line 5: remove empty space after "4.3.1)".

Response: This has been done.

Page 5, line 36: Check sentence (". . . here, data . . .") Page 29, Figure 13: Please improve x-axis numbering so that it is easier to see the beginning of each year.

Response: This has been done.

Pages 14 – 18: References: Please check all references carefully: Needs some harmonization, e.g., with respect to authors list: Currently it is (i) a mix of listing only one author, or several or all (sometimes even with ". . ."), and (ii) author's given name is typically abbreviated but not always, etc. Furthermore mix of "in preparation" and "(in prep)".

Response: This has been done.

---

## Author Comment (AC2) · 23 Dec 2016

From the author: We appreciate the time investment that anonymous referee #2 made to review our paper. We carefully considered all of the comments and have responses to each. We agree with their assessment that the paper is limited in novel science findings, but will serve as a valuable reference, and we are grateful that they understand this about the scope of the paper.

I have some comments/suggestions, mainly related to section 5 that the authors should address. There is also a number of minor corrections that need to be applied.What is the reason of showing the growth rate per latitude as figure 12 seems to show very little latitudinal difference. Can you please discuss what you would expect to find and what you do see from OCO-2. What can we learn from the comparison of column

data from OCO-2 to Manu Loa in-situ data? Both don't agree very well (eg for Sep the difference is almost 1 ppm) so I don't understand if you try to say that OCO-2 data shows a reasonable growth rate or not. At least, you could select only OCO-2 data around Mauna Loa. Clearly, much more useful would be a comparison to the growth rate from OCO-2 data a TCCON station.

Response: We have considered the reviewer's comments, and agree that we just touch on a rich topic here. I have added citations to more complete work on this topic, and restated our message, that for a first look, the OCO-2 growth rates are reasonable – ie, we should look further, and not abandon the project now.

Very similar questions apply to figure 13. This is an 'apples to oranges' comparisons. I think it would be useful to add column data from a model constraint with in-situ data such as CarbonTracker or CAMS so that proper column to column comparisons are possible.

Response: We appreciate this comment. We did some analysis with a modeling group as we developed the paper, but concluded that the model OCO-2 comparison is a large topic that merits a separate publication. In addition, there is a lot of variability from model to model, so including just one model result is not telling the whole story. Both reviewers had comments about this figure, focused on the 'apples to orange' comparison of total columns and surface measurements. It would be significant additional scope to discuss the vertical distributions, averaging kernels, and how to properly compare the surface data to the total columns. Therefore, we have decided to limit this discussion to just OCO-2 data, showing a weekly timeseries and pointing out the standard deviation of the OCO-2 data relative to the changes in time. We have also improved the time axis on the graph and pointed to the TCCON timeseries in the Wunch et al. (2016) paper.

Also, figure 14 does not add much value as no context is given. Adding data from a model to the figure (see above) would allow to gauge if the OCO-2 is roughly consistent

with expectations.

Response: Based on your comments and the comments of another reviewer, we have decided to eliminate Figure 14 from the paper. You are correct in that more detailed are needed to explain what we have learned and what questions remain about the observed gradients, and we will leave that to later publications.

Minor comments:

p.1 : Correct the assignment of affiliations of authors

Response: This has been done.

p.2, l. 9: For mass balance. . . -> For mass balance reasons. . .

Response: This has been done.

p1. L. 34 measure atmospheric CO2 with -> measure atmospheric CO2 columns with

Response: This has been done.

p.5 l. 35: error of 4% will impart an XCO2 error of 0.22 ppm, 0.12 ppm, and 0.4 ppm -> Connor et al., AMT, 2016 seems to suggest a much larger error in CO2 as a result of 4% radiometric uncertainty. How did you calculate this CO2 error?

Response: We added a statement about the sensitivity calculation that was used to derive this.

p.6 (optical depths less than âĹij0.35) -> does this refer to the retrieved optical depth which must not agree with the true optical depth?

Response: This is clarified in the paper – it refers to the OD from the prescreeners.

p.7 SNR design requirements were 290, 270, and 190 at nominal radiance levels -> please give the nominal radiance levels

Response: The nominal radiance levels are now included in the paper.

p.9 eq. 4-1: y, x, F, e are vectors and should be given in bold or use index to show that this is a sum over elements of the vector. Vector should be given in bold in the following paragraph as well. Also, define n.

Response: This has been done.

p. 11 l. 19: selected glint water data only -> selected glint data over water only

Response: This has been done.

p. 11, l.19 175 W to 130W in longitude, and from 15N to 25N -> 175ËĘo W to 130ËĘo W in longitude, and from 15ËĘo N to 25ËĘo N

Response: This has been done.

p. 17, l.5: retrieval -> retrieval

Response: This has been done.

p. 17, l. 5-8: There are 2 references labelled O'Dell et al., 2016 Can you confirm that Fig. 6-9 show the same coverage as Fig.. 3-5. It looks different but this might simply due to the different figure size.

Response: Yes, the reviewer is correct, this is an artifact of the scale of plotting.